# Mapping the function of neuronal ion channels in model and experiment

William F Podlaski[1,2*†], Alexander Seeholzer[3,4,5†], Lukas N Groschner[1,2], Gero Miesenböck[1,2], Rajnish Ranjan[6], Tim P Vogels[1,2]

[1]Centre for Neural Circuits and Behaviour, University of Oxford, Oxford, United Kingdom; [2]Department of Physiology, Anatomy and Genetics, University of Oxford, Oxford, United Kingdom; [3]School of Computer and Communication Sciences, École Polytechnique Fédérale de Lausanne, Lausanne, Switzerland; [4]School of Life Sciences, Ecole Polytechnique Federale de Lausanne, Lausanne, Switzerland; [5]Brain Mind Institute, Ecole Polytechnique Federale de Lausanne, Lausanne, Switzerland; [6]Blue Brain Project, École Polytechnique Fédérale de Lausanne, Geneva, Switzerland

**Abstract** Ion channel models are the building blocks of computational neuron models. Their biological fidelity is therefore crucial for the interpretation of simulations. However, the number of published models, and the lack of standardization, make the comparison of ion channel models with one another and with experimental data difficult. Here, we present a framework for the automated large-scale classification of ion channel models. Using annotated metadata and responses to a set of voltage-clamp protocols, we assigned 2378 models of voltage- and calcium-gated ion channels coded in *NEURON* to 211 clusters. The *IonChannelGenealogy* (ICGenealogy) web interface provides an interactive resource for the categorization of new and existing models and experimental recordings. It enables quantitative comparisons of simulated and/or measured ion channel kinetics, and facilitates field-wide standardization of experimentally-constrained modeling.

*For correspondence: william.podlaski@cncb.ox.ac.uk

†These authors contributed equally to this work

**Competing interests:** The authors declare that no competing interests exist.

## Introduction

Ion channels play crucial roles in neuronal signal processing (*Koch and Segev, 2000*; *Cai et al., 2004*; *Goldberg et al., 2008*) and plasticity (*Sjöström and Nelson, 2002*; *Shah et al., 2010*; *Debanne et al., 2003*). Interactions among the many different ion channels expressed by a single cell can lead to extraordinarily complex dynamics, whose dissection necessitates computational modeling, as first demonstrated by *Hodgkin and Huxley (1952a)* for action potential generation. Simulation environments like *NEURON* (*Hines and Carnevale, 2001*; *Carnevale and Hines, 2006*) can be used to create biophysical neuron models with realistic morphologies, ionic currents, and channel densities (*Figure 1A*), facilitating the integration of experimental data into models (*Mainen and Sejnowski, 1995*; *Stuart and Spruston, 1998*; *Migliore et al., 1999*; *Poirazi et al., 2003*; *Destexhe and Paré, 1999*; *Traub et al., 2003*). More than a thousand neuronal models, and several thousand individual ion channel models, are archived in the online database *ModelDB* (*Hines et al., 2004*), which enables other researchers to verify original claims, and to reuse and extend existing neuron models in the light of new results.

Matching model and experiment is essential for biophysical neuron models, in which many components have a direct biological counterpart (*Brette et al., 2007*). For example, pyramidal neuron models have been shown to reproduce the recorded spiking activity of these cells accurately with a particular set of ion channels (*Traub et al., 2003*; *Figure 1B*, gray traces; see Materials and methods). However, the dynamics can change, sometimes dramatically, when one of the modeled ion

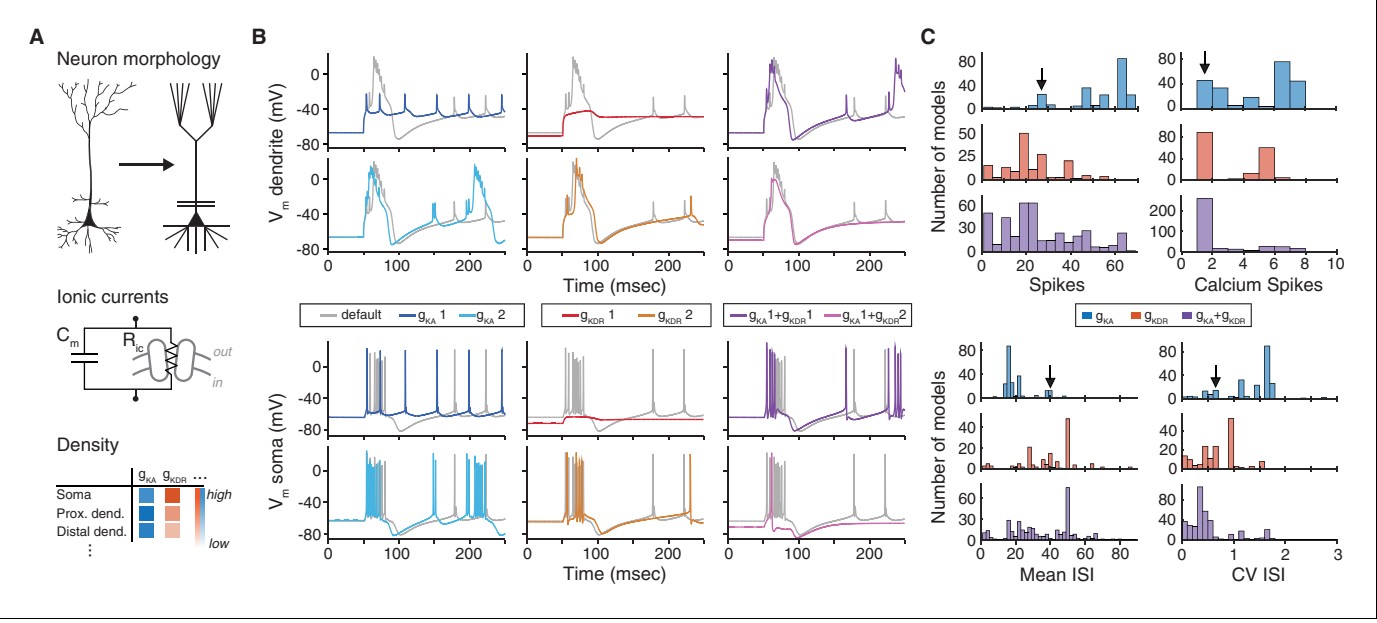

**Figure 1.** The choice of ion channel model influences the behavior of a simulated neuron. (**A**) Biophysical neuron models are composed of a detailed multicompartmental morphology, several active ion channel conductances, and a density of each conductance that depends on the specific compartment. (**B**) Simulation of a detailed layer 2/3 pyramidal neuron model, adapted from *Traub et al. (2003)* (see Materials and methods for details). The neuron model was stimulated with a 1.5 nA current step beginning at 50 ms, while recording the membrane potential in the apical dendrite (top) and soma (bottom). Simulations were first run using the original conductances from Traub et al. ('default', gray). Left: the default A-type potassium model ($g_{KA}$) was replaced with two other A-type models ($g_{KA}1$: dark blue, *Hay et al. [2011]*, ModelDB ID no. 139653; $g_{KA}2$: light blue, *Traub et al. [2005]*, ModelDB ID no. 45539). Middle: the default delayed rectifier potassium model ($g_{KDR}$) was replaced with two other delayed rectifier models ($g_{KDR}1$: red, *Zhou and Hablitz [1996]*, ModelDB ID no. 3660; $g_{KDR}2$: orange, *Durstewitz et al. [2000]*, ModelDB ID no. 82849). Right: both A-type and delayed rectifier models were replaced with other models ($g_{KA}1 + g_{KDR}1$, purple; $g_{KA}1 + g_{KDR}2$, magenta). (**C**) Model from B was simulated for 1000 ms with a 1.5 nA current step. Firstly, the default A-type current model was replaced with each of the 243 A-type-labeled model on ModelDB (blue). Secondly, the default delayed rectified current was replaced with each of the 188 delayed rectifier models on ModelDB (red). Finally, the default A-type and delayed rectifier currents were replaced with a random sample of approximately 1% of all possible combinations of A-type and delayed rectifier models on ModelDB (purple). Summary measures are shown for total number of spikes, total number of calcium spikes, mean inter-spike interval (ISI) and coefficient of variation (CV) of ISI during the 1000 ms period. Black arrows represent the simulation results for the default model.

channel currents is exchanged for an identically-labeled model from a different publication on *ModelDB* (*Figure 1B*, colored traces, *Figure 1C*). This example underscores the importance of selecting ion channel models, yet there is currently no standardized experimental dataset against which to validate them.

Furthermore, the increasingly large number of models on ModelDB (e.g., over 300 new ion channel models in 2014 alone; *Shepherd Lab [2015]*), with non-standardized labeling and a high degree of redundancy, makes it difficult to understand how ion channel models relate to each other and to biology. For example, a researcher looking to use an existing *A-type* potassium channel model will find over 250 *A-type* models, spanning a range of behaviors (*Figure 1C*, blue). Instead of a thorough and time-consuming fitting of appropriate ion channel dynamics, it is common for modelers to adapt previously published ion channel models for their own purposes. However, this may introduce experimentally unverified systematic changes or even errors into later generations of models and may have dramatic effects on the biological interpretation of the results.

To facilitate informed choices among this bewildering variety of ion channel models, we categorized 2378 published voltage- and calcium-dependent ion channel models in *NEURON* that are available on *ModelDB*. We cataloged all relevant information about each ion channel model from the associated literature, including its *pedigree* relations: whether a given ion channel model is based on previous models, and, if so, which ones. Additionally, we compared the kinetics of each ion channel model in standardized voltage-clamp protocols. The resulting maps of ion channel behavior show model variability and diversity, and point to the computational and experimental sources that were

used to fit each model. Our efforts have grouped 2378 ion channel models into 211 clusters, dramatically simplifying the search for an appropriate ion channel model.

We present our findings in an annotated, interactive web-interface with a short video manual (*ICGenealogy, 2016*; https://icg.neurotheory.ox.ac.uk/), that allows filtered search of individual ion channel models by metadata and relational information, and the comparison of channel model kinetics. The underlying database is freely and programmatically accessible via a web application programming interface (API). In an effort to make our resource compatible with experimental data and new ion channel models, we offer the possibility to upload and assess the similarity of experimentally recorded current traces (as well as new models and model traces) in the same topology. We show an example of the use of this comparison through the analysis of an unclassified ion channel model, as well as an experimentally recorded voltage-dependent potassium current from *Drosophila melanogaster*. In summary, we provide a framework for the direct and automated comparison of models and experiments to facilitate experimentally constrained modeling and quantitative characterization of ion channel behavior.

## Results

### Categorizing ion channel models by metadata and ancestor-descendant relationships

To build a map of ion channel model function, we categorized and analyzed a widely-used subset of 2378 voltage- and calcium-dependent ion channel models ('.*mod*' files) in the *NEURON* simulation environment (*Hines and Carnevale, 2001*; *Carnevale and Hines, 2006*). A set of 'metadata' was extracted manually from the associated journal articles for each ion channel model file (*Figure 2A*, top): reference information (Ref. Info, including author(s) of the model code), ion channel information (I.C. Info: ion selectivity, gating mode, subtype), system information (Sys. Info: brain area, neuron type, neuron region, animal model), as well as additional comments (Other: e.g. temperature constraints, see Materials and methods).

Based on ion selectivity and gating mode, the majority of all ion channel models (~75%) fell into five classes (*Figure 2B*): voltage-dependent potassium (Kv), voltage-dependent sodium (Nav), voltage- dependent calcium (Cav), calcium-dependent potassium (KCa) and hyperpolarization-activated cation (Ih) channel models. We recorded 74 different subtype labels across all classes (Kv: 32, Nav: 19, Cav: 20, KCa: 11, Ih: 5; *Figure 2B*, cf. *Figure 2—source data 1*). Prominent modeled neuron types were *pyramidal*, *interneuron*, *granule cell*, and *basket cell* (*Figure 2C*), and prominent brain areas included *hippocampus*, and *cortex* (*Figure 2D*). Other metadata also showed a broad variety across ion channel models (*ICGenealogy, 2016*).

To denote family relations (*Figure 2A*, bottom), ion channel model *A* was labeled as a 'descendant' of an older 'ancestor' ion channel model *B* if the publication reporting *A* cited the publication for *B* as the source or starting-point of its channel dynamics, or if the code of models *A* and *B* were sufficiently similar (Materials and methods). By establishing a citation relationship between different models, we effectively create a *genealogy* of neuronal ion channel models, which describes their lineage (not to be confused with the actual genetic ancestry of different types of ion channels). Visualizing family relationships makes it apparent that many ion channel models form large families, often with a highly-cited hub model that has many descendants (*Figure 2E*). On the other hand, there are a large number of small families and model singletons that imply de novo ion channel model creation, lack of appropriate citations, or translation from other simulators (noted in the metadatum *comments*). Subtype labeling mapped well onto families, but family identity did not guarantee homogeneity of subtype or vice versa – all individual subtypes were found across several families (see *Figure 2E* for Kv, and *Figure 4—figure supplements 1* and *2A,F* for other ion type classes).

Family relationships and metadata thus help to distinguish ion channel models, but the lack of standardized annotations in a common nomenclature, as well as the sheer abundance of models make it difficult to infer the degree of their functional diversity. Based on metadata alone, it is thus difficult to choose an ion channel model for appropriation into one's own work.

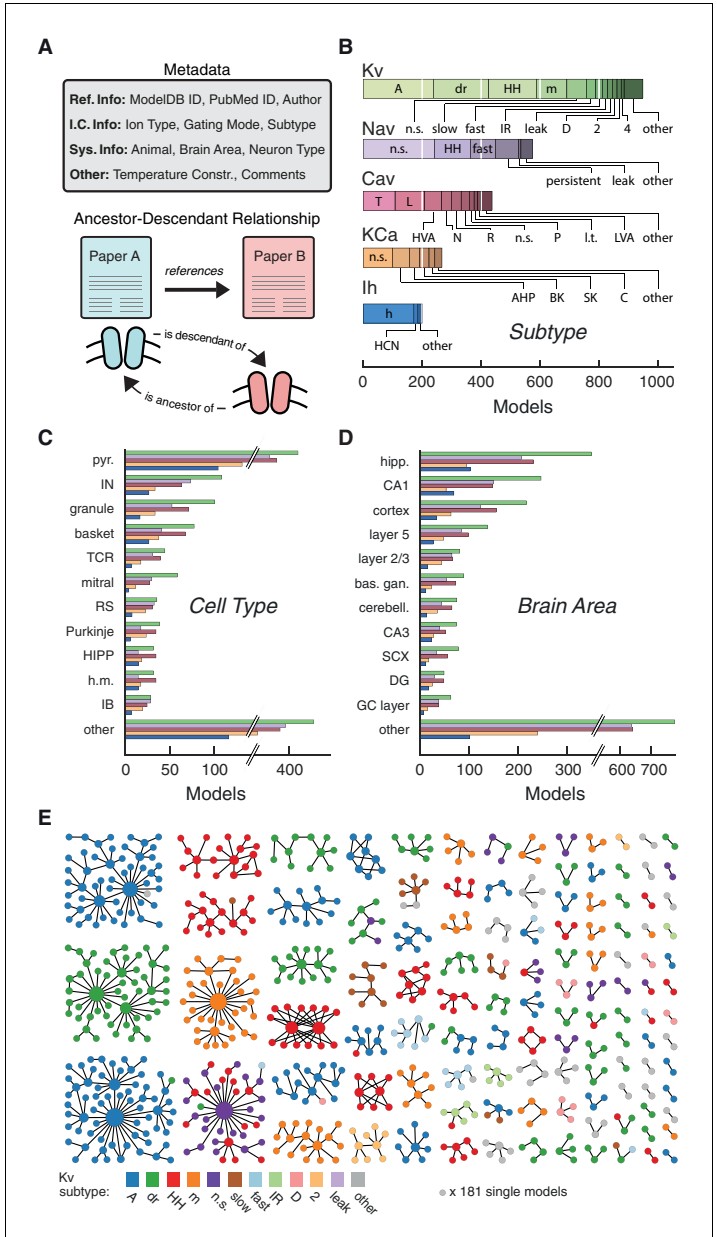

**Figure 2.** Ion channel models can be categorized by metadata and ancestor-descendant relationships. (A) Metadata were manually extracted from ModelDB and associated journal articles (top). Ancestor-descendant relationship (bottom) was established between different models (see main text for description). (B) Models were divided into five classes based on ion type: voltage-dependent potassium (Kv), voltage-dependent sodium (Nav), voltage-dependent calcium (Cav), calcium-dependent potassium (KCa), and hyperpolarization-activated cation (Ih). Each class is divided into subtypes, ordered from left to right according to group size. Uncommon subtypes are grouped together (other). (C, D) Histogram of cell types and brain areas for each ion type, ordered from top to bottom by the number of models. Colors as in B. (E) Pedigree graph displaying families of the Kv class, sorted by family size. Each node represents one model, colored by subtype, and edges represent ancestral relations between models (panel A, bottom). Note that unconnected models (181 total) are not shown. A: A-type, dr: delayed rectifier, HH: Hodgkin-Huxley, m: m-type, n.s.: not specified, IR: inward rectifier, HVA: high-voltage activating, N: N-type, R: R-type, P: P-type, l.t.: low threshold, LVA: low-voltage activating, AHP: after-hyperpolarization, BK: big conductance, SK: small conductance, HCN: Hyperpolarization-activated cyclic nucleotide-gated, pyr.: pyramidal, IN: interneuron, TCR: thalamocortical relay, RS: regular spiking, HIPP: hilar perforant-path associated, h.m.: hilar mossy, IB: intrinsic bursting, hipp: hippocampus, bas. gan.: basal ganglia, cerebell.: cerebellum, SCX: somatosensory cortex, DG: dentate gyrus, GC: granule cell.

*Figure 2 continued on next page*

*Figure 2 continued*

The following source data is available for figure 2:

**Source data 1.** Table of subtypes for each ion type class.

## Defining functional groups of models through voltage clamp protocols and clustering

To quantify the functional relationships between ion channel models, we used a set of voltage-clamp simulation protocols, in kind with those developed for the experimental characterization of ion channels and model fitting (*Hodgkin and Huxley, 1952b*; *Willms et al., 1999*; *Ranjan et al., 2011*). We chose this procedure to assess the spectrum of possible dynamics in a model-free manner, that is, without explicitly taking into account the underlying equations. This allows for the comparison of ion channel models strictly based on their behavior, and, as we discuss later, the direct comparison with experimental data.

Using the *NEURON* simulation environment (*Hines and Carnevale, 2001*; *Carnevale and Hines, 2006*), each ion channel model was placed individually into a model soma and its current responses to five voltage-clamp protocols were recorded (*Figure 3A*, left; see Materials and methods, *Figure 3—figure supplement 1* and *Table 2* for full description). The protocols were designed to probe the gating characteristics of ion channels, that is, activation, deactivation and inactivation, as well as temporal dynamics during voltage ramping and repeated action potentials. Protocol parameters were adjusted for each of the five ion classes separately. Current responses were normalized to remove the dependence on the maximum conductance, and subsampled at particular regions of interest (*Figure 3A*, dashed areas) to obtain a trace of characteristic data points for each protocol (*Figure 3B*, *Figure 3—figure supplement 1F*). Using principal component analysis (PCA) across all traces of a particular ion channel type, we obtained a final $D$-dimensional score for each ion channel model, accounting for at least 99% of the variance across all channel models in each class. The dimensionality $D$ varied between 16 and 29 dimensions for the five classes (Kv: 16, Nav: 21, Cav: 29, KCa: 16, Ih: 16). The Euclidean distance between any two given model scores was termed their 'similarity' (*Figure 3C*, top). Finally, we used Ward's clustering method (*Ward, 1963*) on the model scores to establish an agglomerative hierarchy of ion channel model clusters (*Figure 3C*, bottom).

A suitable number of clusters was obtained through a variety of published cluster indexes (see Materials and methods, *Figure 3—figure supplements 2* and *3*). For the Kv class, this resulted in 60 clusters with distinct responses (*Figure 4*) and small intra-cluster variability (*Figure 3—figure supplement 4*). The other classes divided similarly into 38, 43, 44, and 26 clusters for Nav, Cav, KCa, and Ih, respectively (see *Figure 4—figure supplements 1* and *2*). We named clusters according to the most common label of their members and we denoted the ion channel model closest to the mean score coordinate of each cluster as its reference model. While many clusters are relatively homogeneous in terms of subtype label, there are several that feature a mix of different subtypes (see the section on variability below). Therefore, the subtype label of clusters should be used as a guide for data exploration rather than as a strict classifier.

We found that most ancestor-descendant families fell within one cluster, indicating consistency between the family relations collected from the papers and ion channel model behavior (*Figure 4A–B*, *Figure 4—figure supplements 1* and *2B,C and G,H*). However, a common subtype label did not guarantee a common cluster identity (*Figure 4B*, *Figure 4—figure supplements 1* and *2C,H*). Many models with the same subtype fell into different clusters. For example, the ~250 A-type-labeled Kv ion channel models fell into 14 clusters (although only five clusters comprised over 90% of them, *Figure 4B*). These clusters contained few other subtype labels, suggesting that *A-type* is generally a consistent label for at least five similar, yet distinct kinetic behaviors. Moreover, the similarity between these clusters was generally high (and thus they were plotted within the same vicinity on the wheel of the 'Circos' plot, *Figure 4D*; see also Materials and methods). Other subtype labels across all ion channel types showed similar results (*Figure 4—figure supplements 1* and *2C, H*). Interestingly, for four of the five ion type classes (KCa being the exception), most isolated single-model clusters corresponded to genealogical singletons, supporting the idea that these ion channel models are indeed unique, and do not appear isolated simply due to missing ancestor-descendant

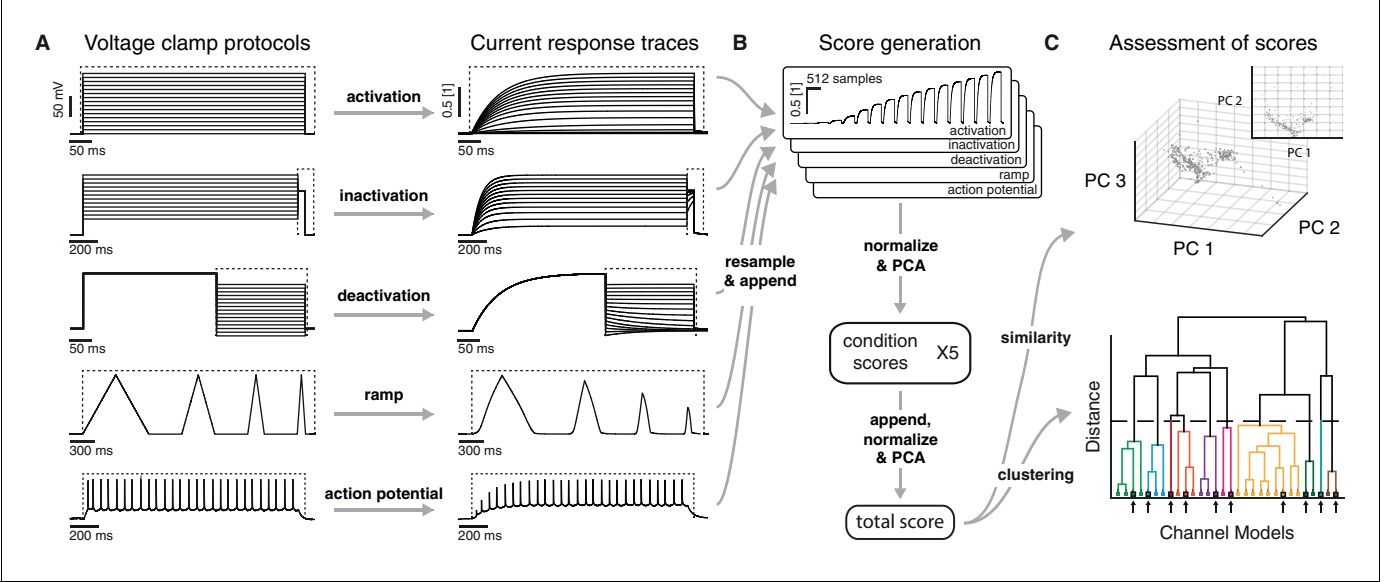

**Figure 3.** Voltage-clamp protocols for the quantitative analysis of ion channel models. (A) Left: five voltage clamp protocols were used to characterize ion channel responses recorded in single compartment somata simulated in *NEURON* (see *Figure 3—figure supplement 1* and *Tables 1* and *2* for full description). Multiple lines indicate a series of increasing voltage steps with the same time sequence. Right: current response traces are shown for an example model. Dashed regions indicate response times used for data analysis. (B) Current responses were subsampled and appended, then dimensionality-reduced by principal component analysis (PCA) to form a *condition score* vector for each protocol. These score vectors were further normalized and dimensionality-reduced to form a *total score* vector. (C) The first three principal components of the score vector are shown for Kv ion channel models (top). Scores were clustered using an agglomerative hierarchical clustering technique (bottom). Distinct clusters (noted by colors) form when a cutoff (dashed line) is introduced in the distance between hierarchical groupings, chosen based on several cluster indexes (see *Figure 3—figure supplements 2* and *3*). *Cluster representative models* (bold squares with arrows) are selected as reference models for each cluster (see Materials and methods).

The following source data and figure supplements are available for figure 3:

**Source data 1.** Table of omitted files.
**Figure supplement 1.** Graphical description of the five voltage-clamp protocols used for ion channel model analysis.
**Figure supplement 2.** Cluster indexes for Kv and Nav classes.
**Figure supplement 3.** Cluster indexes for Cav, KCa and Ih classes.
**Figure supplement 4.** Comparison of intra- and inter-subtype variability with intra- and inter-cluster variability.

links. However, this was not true for all genealogical singletons, many of which were kinetically aligned to larger families and consequently fell into the same clusters. In conclusion, clustering allowed us to identify 211 distinct groups of ion channel models that share similar behavior, regardless of publication context or subtype labeling.

## Ion channel model groups defined by common metadata show variability in behavior

The variability in the behavior of identically-labeled ion channel models in different clusters may stem from various sources. There is substantial evidence that individual neurons of any given type display heterogeneity in ion channel expression and regulation (*Marder and Goaillard, 2006*; *Schulz et al., 2008*). Furthermore, characterizing an ionic current using the average response across a population may not be sufficient to capture the appropriate behavior at the neuronal level (*Golowasch et al., 2002*), as there may be several distinct 'solutions' (*Prinz et al., 2004*). Diversity and variation in ionic currents even within a single cell type may arise from such mechanisms as splice

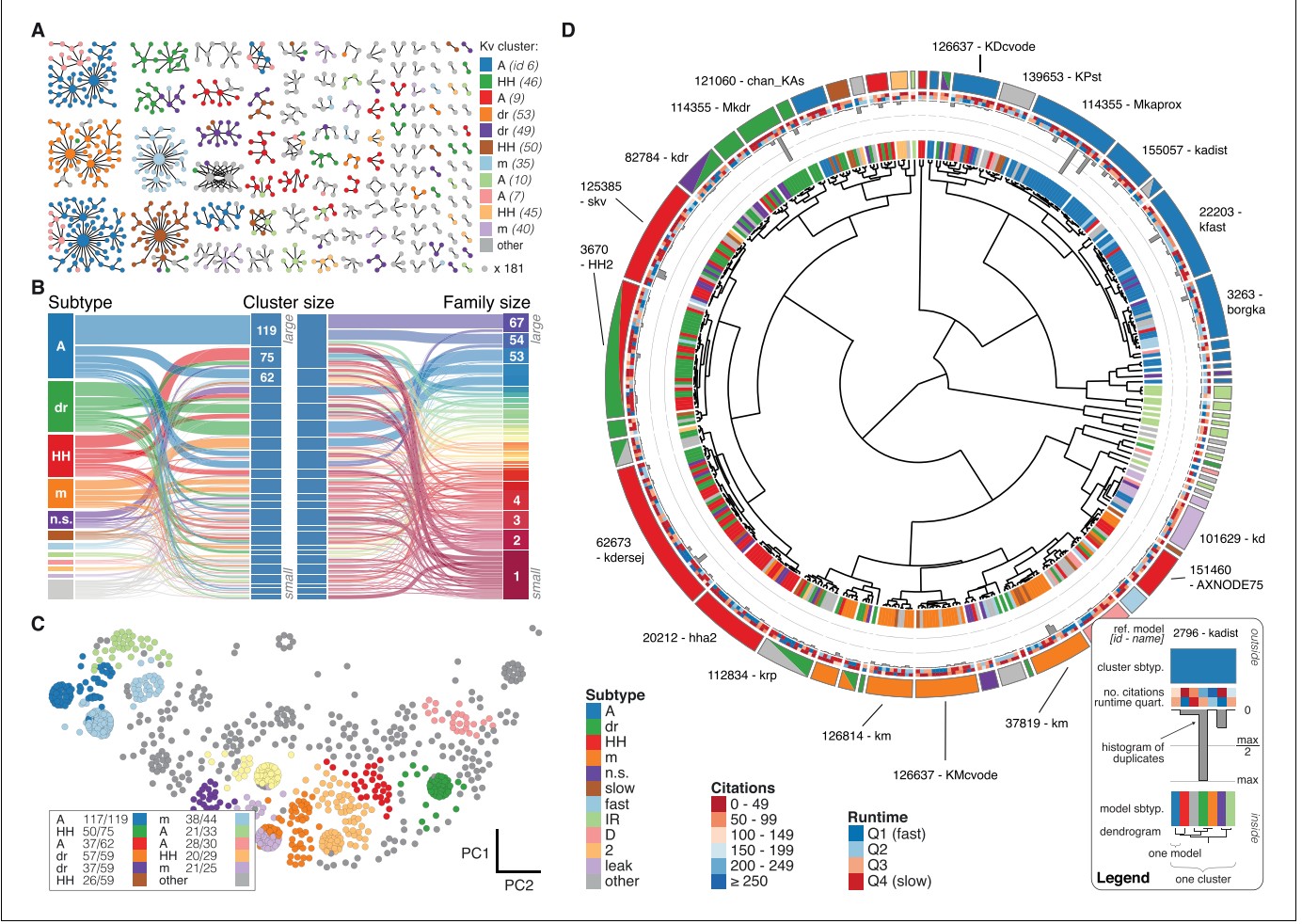

**Figure 4.** Quantitative analysis of Kv ion channel models: functional map and clusters of common behavior. (A) Pedigree graph of the Kv class (cf. *Figure 2E*), colored by membership in the 11 largest clusters in the class (named by most prevalent subtype, bottom). Membership to other clusters is indicated by gray color. Cluster ID is given in parentheses for easy comparison with website. (B) 'Sankey' diagram for the Kv ion type class, showing the relation between subtype, cluster identification and family identification, each ordered from top to bottom by increasing group size. The 11 most common subtypes are shown in color, with all others grouped together in gray. Small families (size 1 to 6 members) are grouped together. (C) Plot of Kv models in the first two principal components of score space. Colors indicate membership in one of the 11 largest clusters in the class, with membership to other clusters colored in gray. Clusters are named by their most common subtype, with the proportion of that subtype specified in the legend. Points lying very close to each other have been distributed around the original coordinate for visualization. (D) 'Circos' diagram of the Kv ion type class. All unique ion channel models are displayed on a ring, organized by cluster identification. From outside to inside, each segment specifies: cluster reference model (only displayed for large clusters), cluster subtype(s) (all subtype labels that contribute at least 30%), number of citations, runtime, number of duplicates, model subtype, as well as a dendrogram of cluster connections (black) and family relations (gray). A: A-type, dr: delayed rectifier, HH: Hodgkin-Huxley, m: m-type, n.s.: not specified, IR: inward rectifier. See *Figure 4—figure supplements 1* and *2* for other ion type classes.

The following figure supplements are available for figure 4:

**Figure supplement 1.** Nav and Cav class genealogy and clustering.

**Figure supplement 2.** KCa and Ih class genealogy and clustering.

variants, differential subunit combination, and post-translational modification like phosphorylation (*Schulz et al., 2008*; *Li et al., 2007*; *Campiglio and Flucher, 2015*; *Levitan, 1994*; *Misonou et al., 2004*).

This biologically variability may also contribute to the diversity in ion channel models that we observe in our database. Consistent with this notion, the behavior of groups of ion channel models

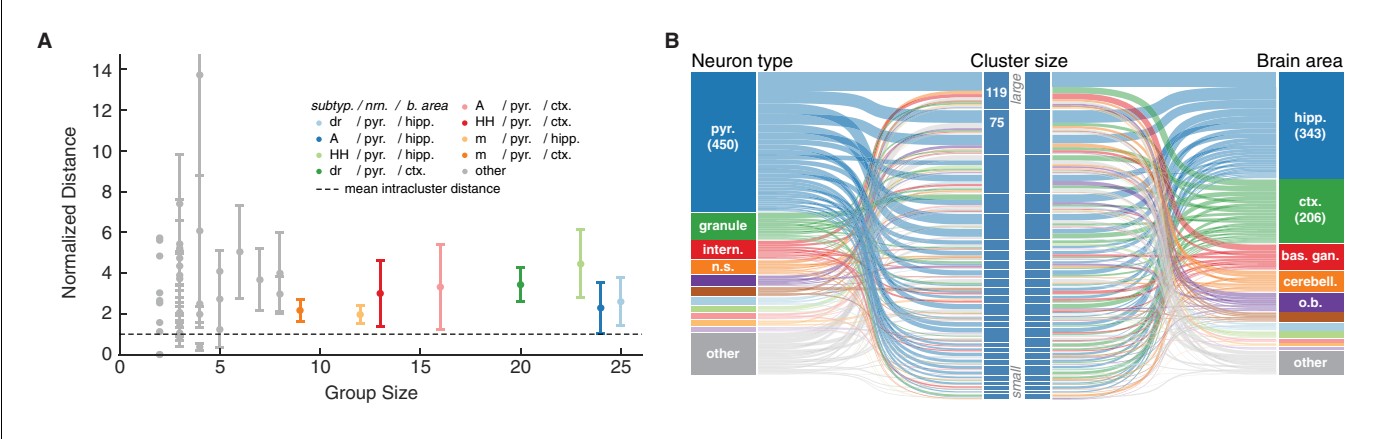

**Figure 5.** Ion channel model groups defined by common subtype, neuron type and brain area show variability in behavior. (**A**) Kv models are grouped by common subtype, neuron type, and brain area. The mean pairwise distance in score space between all models within a group is plotted against group size, with errorbars corresponding to the first and third quartiles (middle 50% of distances). The largest eight groups are shown in color. Distances are normalized relative to the mean pairwise distance between models in the same *cluster* (dashed line; averaged across all clusters with at least one pairwise distance greater than zero, 43 of 60 clusters). Most groups have a larger mean distance than the average cluster (above the dashed line). (**B**) 'Sankey' relational diagram for metadata, neuron type, cluster identification and brain area, for the Kv ion type class. Bars are stacked histograms, that is, the height of the bars indicates the relative number of models. Left column: partition of channels by 11 most prevalent neuron types. Middle column: partition of channels into assigned clusters. Right column: partition of channels by 11 most prevalent brain areas. Links between columns are colored according to neuron type (left) and brain area (right). Numbers in the left/right column are the number of channels per label. pyr: pyramidal, intern.: interneuron, n.s.: not specified, hipp.: hippocampus, ctx: cortex, bas. gan.: basal ganglia, cerebell: cerebellum, o.b.: olfactory bulb.

defined by common subtype, neuron type and brain area (*Figure 5A*, plotted data points) is often more diverse than that of models defined by a common cluster (*Figure 5A*, dashed line). More specifically, we find no clear correspondence of any given cluster with categories such as brain area and neuron type (*Figure 5B*). Nearly every cluster contains ion channel models that have been used in pyramidal cells (*Figure 5B*, left, blue) of both cortex and hippocampus (*Figure 5B*, right, blue and green). In the same vein, e.g., *A-type*-labeled models that have been used in pyramidal cells of the hippocampus (117 models) are found in nine clusters (cf. *ICGenealogy, 2016*).

A portion of the variability may also stem from non-biological sources, such as differences in the experimental setup, as well as model fitting, and idiosyncratic changes to individual ion channel model implementations. Consistently, we find that models defined by common families (connected directly or indirectly through ancestor-descendant relationships) can occasionally fall into different clusters (*Figure 4A*).

It is not possible to disentangle how much of the variability in ion channel kinetics is due to each of these components. While our resource provides, for the first time, a catalogue of all models created for each system, and how they relate to one another, we remain agnostic about the sources of variability seen in the models that we analyze (cf. Discussion).

## Automated comparison of new ion channel models and experimental data

Our analysis framework, accessible through the web interface, enables the automated analysis of new ion channel models as well as experimental data (*Figure 6A*). To illustrate this process, we uploaded and tested a previously uncatalogued Kv model from a hippocampus CA1 pyramidal cell model (*kad.mod* from *Hsu et al. (2015)*; ModelDB ID no. 184054). We compared its scores and response traces to the presently available 931 Kv models (*Figure 6B*) and determined its relation to previous ion channel implementations. We found that the ion channel model fits well within a cluster of mostly pyramidal A-type-labeled ion channel models used in simulations of rodent hippocampus, thereby verifying the assumed characteristics of the model.

The framework can also be used for the comparison of experimental data and models. To illustrate, we uploaded and tested an experimental dataset of recordings from Kenyon cells in

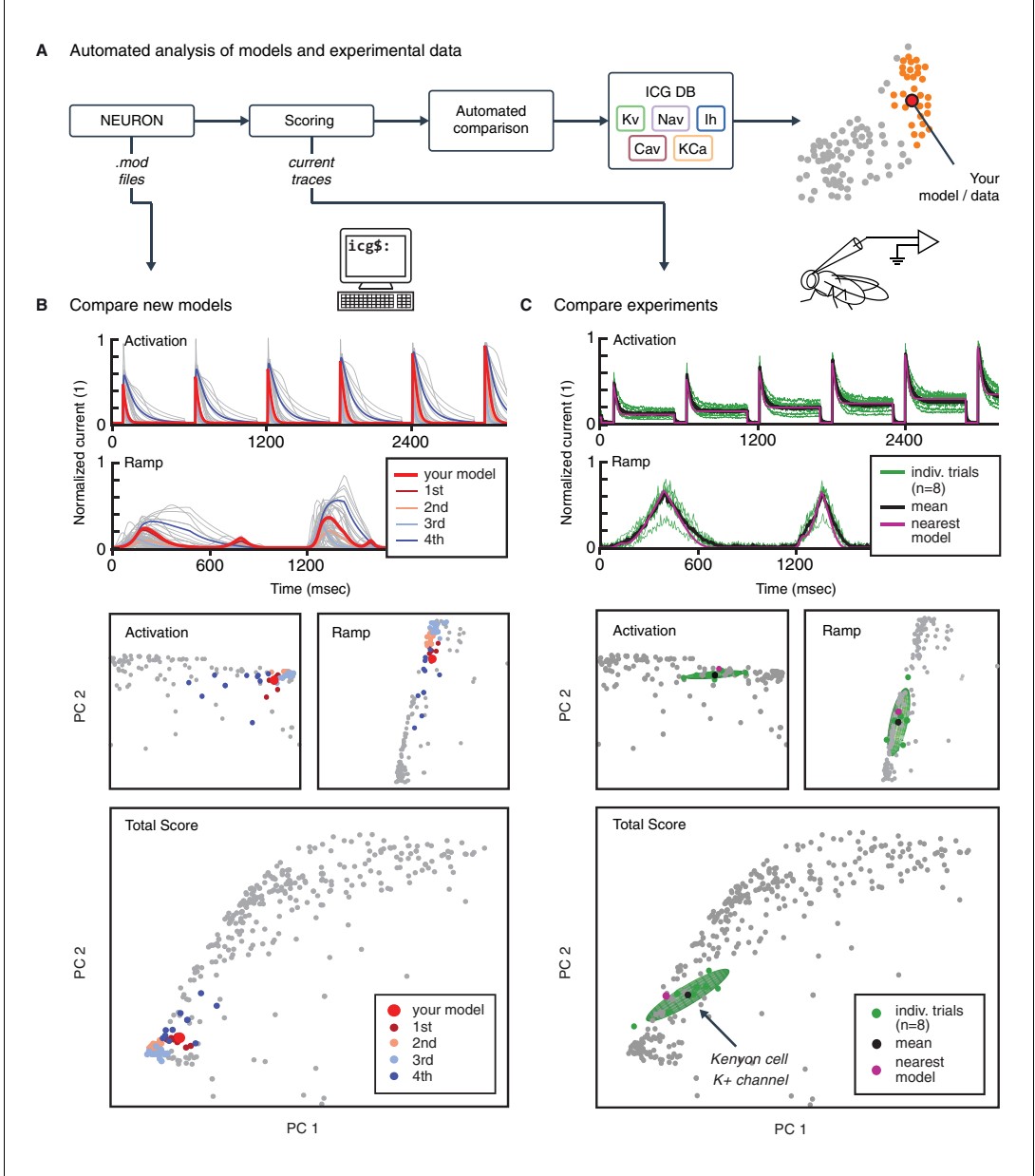

**Figure 6.** Automated analysis of new models and experimental data. (**A**) Flowchart of data processing steps involved in automated comparison. Source code for model files written in *NEURON* can be uploaded to the website, and current responses are automatically generated. Current traces are processed to compute scores, which are compared to all models in the resource (illustrated in **B**). Additionally, raw current traces obtained experimentally (or from models in other languages) can be uploaded and analyzed directly (illustrated in **C**). (**B**) Example analysis and comparison of a new ion channel model (*kad.mod* from **Hsu et al. (2015)**; ModelDB ID no. 184054). Top: Segments of the current response traces (red) for activation (voltage steps 10–60 mV) and ramp protocols (first half), along with the closest four clusters (other colors: mean currents, gray lines: individual currents). Bottom: first two principal components of score space for activation and ramp protocols, as well as total score. (**C**) Example analysis of in vivo recordings of a K+ current from Drosophila Kenyon cells (see Materials and methods for details and *Figure 6—figure supplement 1* for full traces) and comparison to ICG resource. Top: mean (n = 8 recordings, black) and individual recordings (green). Bottom: mean (black dot) and individual experimental recordings (green dots) plotted in the first two principal components of score space (ellipsoid illustrates the variance across individual recordings). Comparison is made to the nearest (in score space) ion channel model in the resource (magenta; *Kv4_csi*, ModelDB ID no. 145672). Gray dots in **B** and **C** are scores of Kv channel models in the resource.

The following figure supplement is available for figure 6:

**Figure supplement 1.** K+ current recordings from Drosophila Kenyon cells.

*Drosophila melanogaster*. Voltage-gated cationic currents across the membranes of these neurons are thought to be dominated by A-type K+ channels, in particular Shal/Kv4 (*Gasque et al., 2005*). This renders Kenyon cells a suitable neuronal cell type to test the biological relevance of our voltage-clamp protocols in an in vivo setting. Current responses were recorded in targeted whole-cell patch clamp experiments in vivo (see Materials and methods). The recordings were performed using our five standardized voltage-clamp protocols, allowing us to transform and compare the experiments directly to ion channel models in the same 'score' space (*Figure 6C*). The comparison revealed a close match to an existing model from our resource, and thus characterizes the behavior of the ion channel as similar to a mammalian Kv4 ion channel (*Figure 6C*; *Fineberg et al. (2012)*; *Kv4_csi*, ModelDB ID no. 145672).

## Discussion

Neuroinformatics has become an increasingly important part of neuroscience research, as new technology and large-scale research projects push the field into the realm of *big data* (*Akil et al., 2011*; *Ferguson et al., 2014*; *Grillner, 2014*; *Tripathy et al., 2014*). Importantly, the need for assessment and aggregation of published knowledge extends beyond experimental data, and has recently started encompassing computational models of neural function (*Hines et al., 2004*; *Gleeson et al., 2013*). Here, we have performed a meta-analysis of voltage- and calcium-dependent ion channel models coded in the *NEURON* programming language available in the database *ModelDB* (*Hines et al., 2004*). Our approach of combining metadata extracted from publications with a kinetics-based analysis allowed us to provide detailed information regarding the identity of each ion channel model in the resource, filling in missing or ambiguous data, and validating the functional properties of channel models against their sometimes ambiguous nomenclature. Furthermore, we provide a framework for the large-scale comparison of models, both with each other and with experiments using the same standardized protocols, thus paving the way towards a unified characterization of ion channel function.

The voltage-clamp protocols used in this study were designed to efficiently probe the kinetics of all ion channel types considered here (*Hodgkin and Huxley, 1952b*; *Willms et al., 1999*; *Ranjan et al., 2011*). Measuring the kinetic responses of each channel model allowed us to compare models regardless of their specific implementation. Notably, our method is amenable to the addition of other protocols that may be better suited to separate certain models. However, there is evidence that simple step and ramp current pulses are sufficient to probe the underlying kinetics of neurons (*Druckmann et al., 2011*), similar to the voltage-clamp protocols that we use here. Additionally, the simplicity of our protocols makes experimental comparison easier.

Additionally, our study can be extended beyond the selection of ion channel models considered here. We limited our analysis to voltage-dependent and calcium-dependent ion channel models coded in the *NEURON* language, but, given the appropriate protocols, other types of ion-channels can be included. The same protocols can also be used to integrate models written for other simulators, or even simulator-independent formats, e.g., NeuroML (*Cannon et al., 2014*). We have taken steps to integrate our resource and visualizations tightly with existing online resources, notably *ModelDB* (*Hines et al., 2004*).

The end result of our work is a dramatically reduced group of candidate ion channel models to test when looking for particular ion channel dynamics. Of the 2378 models in our resource, we could identify 1132 models as unique, and further reduced this to 211 groups with substantially different kinetics. However, this does not eliminate the task of finding the most appropriate ion channel model current, and we stress that the partition of channel models into clusters of similar response properties does not imply that models in the same cluster are necessarily redundant. Clustering is not trivial, and while we have used several measures to determine an appropriate partition of models, we cannot escape a certain level of ambiguity. Intra-cluster differences may still be important depending on the particular simulation at hand. Since the responses of different channels vary slowly and continuously rather than in discrete steps along the dimensions of the manifold of scores (see e.g. *Figure 4B*), the data may also be amenable to more sophisticated clustering and machine-learning approaches. To this end, the raw response data and scores have been made publicly available (*ICGenealogy, 2016*).

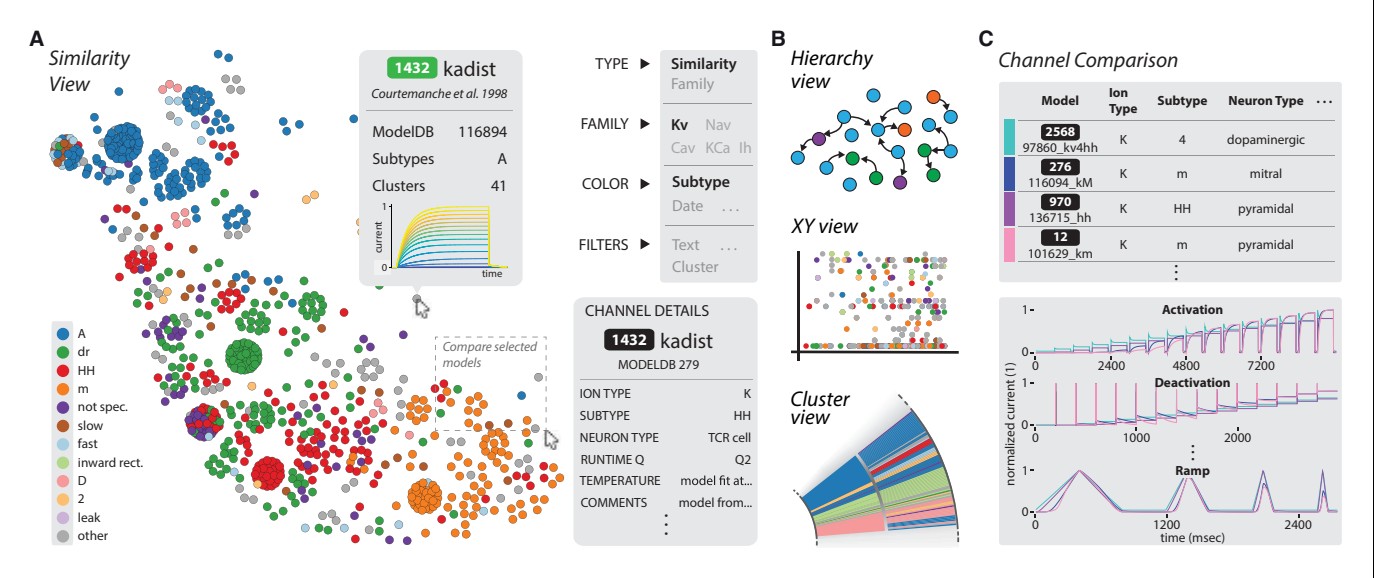

**Figure 7.** The ICGenealogy website allows for the interactive visualization of all data and analysis on the resource (*ICGenealogy, 2016*) (http://icg. neurotheory.ox.ac.uk). (**A**) Schematic of similarity view on website. Channel models of the Kv family are displayed in the first two principal components of score space, colored by subtype (legend on left). Hovering over models brings up information tooltip (center), and clicking on a model displays Channel Details (bottom right). Selected models can be compared by click-and-drag (see instruction manual in *Supplementary file 1* for more details). (**B**) Schematic of other three views available on the website. Hierarchy view: models are displayed in a graph with edges that represent family relations. XY view: any two selected metadata are plotted against each other. Cluster view: models are organized in a ring partitioned by clusters. (**C**) The channel comparison displays selected channels side-by-side with metadata (top) and current responses (bottom).

Furthermore, the variability in behavior of ionic currents seen here and elsewhere (*Marder and Goaillard, 2006*) suggests that there is no clear answer to the question of which ion channel model (and parameters) to use for a given neuronal simulation at hand. However, for modellers who would like to be more diligent about the sources of ion channel models and the comparability across models of the same underlying biological phenomenon, our resource takes a step in this direction.

On a larger scope, it has been suggested previously that neuron model parameters should be viewed as regions, rather than as individual representative points in parameter space (*Goldman et al., 2001*). It may be possible that the variability seen at the level of individual ion channel models covers such regions, and organizes into a handful of distinct 'solutions' to particular model behaviors on the neuronal scale (*O'Leary et al., 2014*). In this sense, our database would lend itself to systematic analyses of the co-variability of sets of published ion channel models that are able to elicit desired behaviors in conductance-based neuron models (see also *Figure 1*), in line with previous work (*Prinz et al., 2003*).

We provide an interactive browser (*ICGenealogy, 2016*), which acts as a complement to existing resources such as ModelDB. It allows the comparison of channel models in five views: a similarity view focusing on the channel's response kinetic scores (*Figure 7A*), a hierarchical tree view focusing on genealogical data (*Figure 7B*, top), an XY view to sort data by a given set of metadata dimensions (*Figure 7B*, middle), and a circular cluster view (*Figure 7B*, bottom). All these views feed a central comparison tool (*Figure 7C*), in which the metadata and traces for user-selected channel models can be viewed side-by-side. For specific examples of how to utilize this browser and to search for specific ion channel models, please refer to the instruction video (*ICGenealogy, 2016*) and manual (*Supplementary file 1*).

Because our voltage-clamp protocols are inspired by experimental procedures, ion channel models can be compared directly to experiments in an automated fashion. We have taken the first steps in this direction by showing a comparison of both a new model and an experimental dataset to the resource here (*Figure 6*). While it is beyond the scope of the current study to integrate ion channel information from the IUPAR/BPS Guide to Pharmacology (*Pawson et al., 2014*), Channelpedia

(*Ranjan et al., 2011*) or other sources, these are important future steps which would help standardize nomenclature. Beyond its usefulness for cataloguing ion channel model behavior and pedigree, our resource will enable better experimentally-constrained modeling, and presents a first step towards a unified functional map of ion channel dynamics in model and experiment.

## Methods and materials

### Pyramidal cell model (*Figure 1*)

A model of a layer 2/3 pyramidal cell was adapted from a previous study (*Traub et al., 2003*), ModelDB ID no. 20756. It contained 68 soma-dendritic compartments and six axonal compartments, with the following active conductances: leak ($g_L$), transient (inactivating) Na+ ($g_{NaF}$), persistent (noninactivating) Na+ ($g_{NaP}$), delayed rectifier K+ ($g_{KDR}$), transient inactivating K+ ($g_{KA}$), slowly activating and inactivating K+ ($g_{K2}$), muscarinic receptor-suppressed K+ ($g_{KM}$), fast voltage- and calcium-dependent K+ ($g_{KC}$), a slow calcium-dependent K+ ($g_{KAHP}$), low-threshold inactivating Ca2+ ($g_{CaT}$), high-threshold non-inactivating Ca2+ ($g_{CaH}$), hyperpolarization-activating cation conductance ($g_{Ih}$). We refer the reader to *Traub et al. (2003)* for channel kinetics, distribution and other details of the model.

The neuron model was simulated in the *NEURON* simulation environment (*Hines and Carnevale, 2001*; *Carnevale and Hines, 2006*), with a current step input injected into the apical dendrite, following Figure 2 of *Traub et al. (2003)*. The protocol was as follows: 400 ms at −0.15 nA, followed by 1000 msec at 1.5 nA. A subset of this trace is shown in *Figure 1B*, comprising 50 ms at −0.15 nA and the first 200 msec at 1.5 nA. The gray traces in *Figure 1B* show the default behavior of the neuron model in response to injected input. The following four measures were computed for each spike train: total number of spikes, total number of calcium spikes, mean inter-spike interval (ISI) and coefficient of variation of ISI.

The stimulation paradigm was repeated in the presence of alternate ion channel models for $g_{KA}$, taken from ModelDB (243 models total). All other ionic conductances, parameters and distributions remained the same. This was further done with alternate ion channel models for $g_{KDR}$ in a separate simulation (188 models total), with $g_{KA}$ set back to the original model. Finally, this was done in the case of replacing both $g_{KA}$ and $g_{KDR}$ currents. A random subset of approximately 1% (441 of the total 45684) of pairs of alternate ion channel models were run together.

### The *ModelDB* database, *NEURON* language and nomenclature

The *ModelDB* database archives published neuron and network models (*Hines et al., 2004*). It contains over 1000 entries, with thousands of ion channel models. At the time of analysis, 496 of the entries were implemented in the *NEURON* language (*Carnevale and Hines, 2006*; *Hines and Carnevale, 2001*), making it the most used simulation environment on the database. Customizable ion channel models are coded in *NEURON* in so-called *.mod* files (with suffix '.mod') (*Hines and Carnevale, 2000*). Mod files for all *NEURON* entries were downloaded from the *ModelDB* website. Each mod file was renamed by adding the *ModelDB* ID as a prefix to the file, in the following way: *ID_name.mod*, where *ID* is the *ModelDB* ID, and *name* is the original name of the *.mod* file. 46 *.mod* files contained more than one current of the same ion type, and were separated into distinct files for each one. The suffix 'icgXY' was appended to the name, where X was the ion type, and Y was the number of the current, beginning with 1 (e.g., 1234_kv_icgK2.mod for the second Kv current in file 1234_kv.mod). Furthermore, some ModelDB entries contained more than one file with the same name. These files were added separately and given unique names by appending a version number to the name – e.g. '_v2' for the second file. The total number of files collected from the database was 3495.

### Collection of metadata

Metadata information was collected using all information in journal articles and files associated with each *ModelDB* entry of interest (SOM). Each field is listed below and defined. Note that some channel models may have missing entries for information that was not stated explicitly in the journal articles or *ModelDB*. Further, we stress that metadata items corresponding to the intended neuron type, brain area and animal are strictly associated with the modeling context, and are not necessarily representative of the experimental ion channels found in that particular neuron, brain area or animal.

- **ModelDB ID**. Identification number associated with each entry on *ModelDB*. All channels from the same entry have the same number.
- **PubMed ID**. The PubMed citation ID of journal articles associated with this channel's *ModelDB* entry; may contain multiple elements, and can be empty for a select few *ModelDB* entries for which no articles were found on PubMed.
- **ion type**. The ion type, or permeability, of the channel model, as listed in the journal article and .*mod* file. The following ion types were analyzed: potassium (K), sodium (Na), calcium (Ca), nonspecific (NS). If models contained more than one current, all ion types were recorded separately. Other ion types were registered but not included in the analysis.
- **gating mode/mechanism**. The dynamic simulation variable that modulates the kinetics of the model, such as voltage (v), calcium (ca), voltage and calcium (v/ca), sodium (na), chloride (cl), light (o), and g-protein coupled (g). Only v channel models were included in the analysis, with the exception of ca and v/ca models exclusively for the K ion type.
- **subtype**. The listed ion channel type, as detailed in the journal article or the .*mod* file. Subtypes were listed as mentioned without conformation to any naming convention, e.g., (*Ashburner et al., 2000*; *Yu and Catterall, 2004*). If no subtype was given, then the subtype was recorded as *not specified*. A full list of all recorded subtypes is found in *Figure 2—source data 1*.
- **author**. Listed author(s) of the .*mod* file (programmers). If authors were not specified in the .*mod* file or on *ModelDB*, we recorded the field as *not specified*.
- **animal**. The animal model (and age, if specified) emulated in simulations, either stated explicitly, or inferred from the journal article.
- **brain area/layer**. The emulated brain area and layer of the simulation, as stated explicitly, or inferred from the journal article.
- **neuron type**. The emulated neuron type of the simulation. May be several types, or listed as *general* if no neuron type was specified.
- **neuron region**. The neuron region that the ion channel is found in, divided into dendrites, soma, axon, axon hillock, or specific areas of dendrites or axon.
- **comments**. Comments from the .*mod* file itself and any other information about the channel and model from the journal article, such as previous models or experimental data that were used to constrain the model.
- **runtime [ms]**. Elapsed CPU time for running 10 repetitions of a single voltage-clamp protocol (action-potential). In plots and on the web interface, we simplify model runtimes by assignment to one of four quartiles of the distribution of runtimes of all models in each class.
- **temperature**. Details about the model's temperature dependence, and also the temperature at which simulations and/or experiments were performed as described in the journal article.
- **citations**. Estimated number of citations as available through Google Scholar, scraped monthly to update the entries.

A total of 3495 .*mod* files were collected from *ModelDB*. 366 of these files were tools, full neuron models, or other items that do not function as ion channel models. Out of the remaining 3150 files, .*mod* files were placed into one of five groups: voltage-dependent potassium (Kv), voltage-dependent sodium (Nav), voltage-dependent calcium (Cav), calcium-dependent potassium (KCa), and hyperpolarization-activated cation (Ih). The calcium-dependent potassium group contained both ca and v/ca channel models without any distinction. These five groups accounted for 2378 files, references for which are available in *Supplementary file 2*. Mod files that did not fit this description were omitted from the analysis. This included pumps and active dynamics (290), receptor models (370), and models with other gating dependencies or ion types (68).

We note that the voltage-dependence of ion channels may stem from different underlying mechanisms. This includes traditional voltage-gated ion channels that contain a voltage-sensitive domain (*Catterall, 1995*), as well as dependence that occurs indirectly through interaction with other molecules such as polyamines and magnesium (*Nichols and Lopatin, 1997*), or intracellular signalling cascades (*Kase and Imoto, 2012*). The ion channel models considered here are often agnostic to the biophysical mechanism, and different types of models can be used to model an arbitrary voltage-conductance relationship (*Destexhe and Huguenard, 2000*). Therefore, voltage-dependence as discussed in this work refers to the functional relationship between voltage and channel conductance, and does not generally depend on any particular biological mechanism.

## Ancestor-descendant relationships

The genealogy of ion channel models was defined by an ancestor-descendant relationship. Each channel model was linked to previous models if a relationship was listed in the journal article. We denoted the groups of models connected by these relations as 'families'. This relation could be specific, along the lines of 'the A channel model's kinetics were adapted from the B channel model in a previous journal article'. Other times, the description was vague, e.g., stating that the neuron model was adapted from a previous one, with no explicit reference to ion channel kinetics. When no information was listed about previous kinetics, both in the journal article and model files themselves, the channels were assumed to have no ancestors and to have been created de novo. However, in many situations obvious similarity in .*mod* file code as verified by a diff command was sufficient to link models to previous ones. In these cases, relations were established even when they were not stated in the journal articles. This task was done by hand, and as such is prone to mistakes. We repeated the collection of metadata, including ancestry relations, a second time in order to correct for potential errors – we hope to correct any remaining missing or superfluous ancestral relations with the help of user submissions (*ICGenealogy, 2016*).

We note that the use of the word *genealogy* in this work is used exclusively to describe the ancestry of ion channel *models*. It does not refer to the genetic lineage of ion channels as found in biology. Furthermore, this genealogy does not necessarily conform to a pedigree as defined by journal article reference information, as sometimes ion channel models are combined from several papers, or references may be missing.

## Voltage-clamp protocol

Mod files were run individually in a *NEURON* simulation by generating a single soma compartment of length and diameter equal to $20\,\mu m$ and cytoplasmic resistivity of $150\,\Omega cm$. A passive conductance was set to $3.334 \cdot 10^{-5} S/cm^2$. The simulation temperature was set to $37\,°C$. Reversal potentials were specified separately for each ion type. Some models (172 files) featured explicit calculation of the reversal potential, so internal and external concentration values were added as extra variables to make these equivalent. Parameter values for reversal potential and ion concentrations can be found in *Table 1*.

A particular model was placed in the soma and a series of five voltage-clamp experiments were run (*Figure 3—figure supplements 1*; *Table 2*), with the current output being recorded. Based on the desired effect of each protocol, only particular sections of the protocols were used in comparing the kinetics (*Figure 3—figure supplement 1*, dashed lines; also noted in *Table 2*). The activation protocol featured a single voltage step level, meant to capture the activation kinetics of the model. The inactivation protocol featured a varying voltage step for a long duration, followed by a second fixed voltage step, measuring the inactivation due to the first step. The deactivation protocol featured a single voltage step at a high voltage, followed by a second voltage step of varying amplitude, meant to measure the deactivation kinetics that occur as the voltage is changed from one level to the other. The ramp protocol featured a series of four up and down ramping voltages, at different slopes. Finally, the action potential protocol features voltage deflections as recorded from the soma of a neuron exhibiting a regular spiking pattern. This was recorded from a L2/3 pyramidal neuron of P14 rat somatosensory cortex (R Ranjan, unpublished). Each of the five ion type groups featured different voltage values and durations based on differences in time constants, reversal potentials, and voltage ranges at which each ion channel class is known to be active – no quantitative comparisons were made between classes. Additionally, calcium gated channel models were simulated at seven different calcium concentrations based on known concentrations (*Neher and Sakaba, 2008*). Values were expressed in concentration as $10^{-x}mM$, with $x$ taking the following values: $2.0, 2.5, 3.0, 3.5, 4.0, 4.5, 5.0$. Voltage-clamp protocols are available for download from the ICG website (*ICGenealogy, 2016*).

A substantial number of .*mod* files (952 files) had to be slightly modified to work with the procedure, in one of the following ways: (1) reversal potential was renamed and made a global variable to be accessed from .*hoc* file, (2) NONSPECIFIC CURRENT was changed to a USEION statement with the correct ion type (3) extra functions and/or data were included through .*inc* files, data tables or extra .*mod* files (4) max conductance was made nonzero (arbitrarily set to 1.0) (5) file was split into multiple files for each current present (6) POINTER variables were removed (7) internal temperature

initialization was removed, and temperature dependence was set to use the global variable 'celsius'. The changed files will be made available upon publication.

All ion channel models were taken from the *ModelDB* repository, as published, with small changes as noted above. This included the assumption that model parameters as chosen by the authors were set to reproduce a given, desired, dynamical behavior which matches experimental data or other constraints. We did not consider changing the internal parameters of given models – this would prevent any feasible comparison of the models, since most given models would be able to generate a large variety of behaviors under changing parameter settings (data not shown).

A small number of .*mod* files were omitted from the analysis due to problems with running the simulation protocols (44 files, *Figure 3—source data 1*). These included files that did not compile for unknown reasons, and files that produced abnormal oscillations, or extreme values. The count of these files was 16 for Kv, 20 for Nav, 5 for Cav, 2 for KCa, and 1 for Ih.

## Data extraction and processing

The most recent version at the time of writing, version 7.3, of the *NEURON* language (*Carnevale and Hines, 2014*) was used to run the simulation protocols. All simulations were run in Ubuntu 12.04.5 on a single core of a Intel Core i7 @ 2.67 GHz with 24 Gigabytes of RAM. *NEURON* models were injected with the five different voltage clamp protocols described above and integrated at a timestep of $dt = 5e^{-2} ms$. The resulting current traces were processed in the following manner. First, inward currents (represented as negative deflections from baseline) were flipped by multiplying the entire trace by $-1$. Next, all current traces were normalized by dividing by the maximum trace value in order to make the result invariant to the max conductance parameter (e.g., $\bar{g}$ in Hodgkin-Huxley sodium current $I_{Na} = \bar{g} m^3 h (V_m - E_{Na})$). We reasoned that the maximum current amplitude depends on the number of channel models in a particular area and was thus not related to the kinetic behavior of the channels. The traces were then subsampled at a resolution of 512 data points within the regions of interest stated above. Finally, for protocols containing graded steps (activation, inactivation, deactivation) the subsampled responses across all $c$ graded voltage steps were appended into one representative vector of length $L = 512 * c$. For calcium gated channels, we performed a similar procedure for each of the $k$ calcium concentrations separately, and then appended them into one representative vector of length $L = 512 * k * c$. See *Figure 3—figure supplement 1F* for a schematic of this process.

## Similarity measure

To remove the time dependence of current response waveforms, we performed discrete principal component analysis (PCA)(*Ramsay and Silverman, 2005*) across the temporal dimension, similar to approaches in spike sorting (*Lewicki, 1998*). To this end, the subsampled and appended current responses for each protocol across all $N$ channels in a family yielded a $NxL$ dimensional data matrix, in which we normalized each column by Z-scoring: we subtracted its mean and then divided by its standard deviation. This matrix was then dimensionality reduced by PCA across the $L$ temporal entries, where we chose the reduced dimensionality to capture 99% of the variability. To normalize the range of scores across conditions while keeping the covariance structure, we divided the score vector of each protocol by the standard deviation of all score entries of this protocol. These normalized scores (denoted by *condition scores*) were then combined into a final score vector. Further

**Table 1.** Parameters for reversal potential and inside and outside concentrations used in simulation protocols for five ion type classes. Ionic concentrations were not used for Ih currents.

|  | $E_{rev}$ (mV) | $[ion]_{in}$ (mM) | $[ion]_{out}$ (mM) |
|---|---|---|---|
| Kv | −86.7 | 85.0 | 3.3152396 |
| Nav | 50.0 | 21.0 | 136.3753955 |
| Cav | 135.0 | 8.1929e-5 | 2.0 |
| KCa | −86.7 | 85.0 | 3.3152396 |
| Ih | −45.0 | — | — |

**Table 2.** Voltage-clamp protocol parameters for the five ion type classes. Times are stated in units of ms, voltages in units of mV. See **Figure 3—figure supplement 1** for graphical description. Items $T_A^*$ and $T_B^*$ represent the starting and ending times, respectively, of the regions used for analysis (dashed areas in **Figure 3** of the main text, as well as **Figure 3—figure supplement 1**).

| | Ion type | $V_0$ | $V_1$ | $V_2$ | $\Delta V$ | $T_1$ | $T_2$ | $T_3$ | $T_A^*$ | $T_B^*$ | | | |
|---|---|---|---|---|---|---|---|---|---|---|---|---|---|
| | Kv | −80 | −80 | 70 | 10 | 100 | 500 | 100 | 100 | 700 | | | |
| | Nav | −80 | −80 | 70 | 10 | 20 | 50 | 30 | 18 | 100 | | | |
| | Cav | −80 | −80 | 70 | 10 | 100 | 500 | 100 | 98 | 700 | | | |
| | KCa | −80 | −80 | 70 | 10 | 100 | 500 | 100 | 95 | 605 | | | |
| **Act** | Ih | −40 | −150 | 0 | 10 | 100 | 2000 | 100 | 95 | 2105 | | | |
| Inact | Ion Type | $V_0$ | $V_1$ | $V_2$ | $V_3$ | $\Delta V$ | $T_1$ | $T_2$ | $T_3$ | $T_4$ | $T_A^*$ | $T_B^*$ | |
| | Kv | −80 | −40 | 70 | 30 | 10 | 100 | 1500 | 50 | 100 | 1600 | 1700 | |
| | Nav | −80 | −40 | 70 | 30 | 10 | 100 | 1500 | 50 | 100 | 1580 | 1750 | |
| | Cav | −80 | −40 | 70 | 30 | 10 | 100 | 1500 | 50 | 100 | 1580 | 1750 | |
| | KCa | −80 | −40 | 70 | 30 | 10 | 100 | 1500 | 50 | 100 | 1595 | 1700 | |
| | Ih | −40 | −150 | −40 | −120 | 10 | 100 | 1000 | 300 | 100 | 1095 | 1405 | |
| Deact | Ion Type | $V_0$ | $V_1$ | $V_2$ | $V_3$ | $\Delta V$ | $T_1$ | $T_2$ | $T_3$ | $T_4$ | $T_A^*$ | $T_B^*$ | |
| | Kv | −80 | 70 | −100 | 40 | 10 | 100 | 300 | 200 | 100 | 400 | 600 | |
| | Nav | −80 | 70 | −100 | 40 | 10 | 20 | 10 | 30 | 20 | 29 | 80 | |
| | Cav | −80 | 70 | −100 | 40 | 10 | 100 | 300 | 200 | 100 | 380 | 700 | |
| | KCa | −80 | 70 | −100 | 40 | 10 | 100 | 300 | 200 | 100 | 395 | 605 | |
| | Ih | −40 | −140 | −110 | 0 | 10 | 100 | 1500 | 500 | 400 | 1595 | 2105 | |
| Ramp | Ion Type | $V_0$ | $V_1$ | $T_1$ | $T_2$ | $T_3$ | $T_4$ | $T_5$ | $T_6$ | $T_7$ | $T_8$ | $T_9$ | $T_A^*$ / $T_B^*$ |
| | Kv | −80 | 70 | 100 | 800 | 400 | 400 | 400 | 200 | 400 | 100 | 100 | 100 / 2800 |
| | Nav | −80 | 70 | 100 | 800 | 400 | 400 | 400 | 200 | 400 | 100 | 100 | 98 / 2800 |
| | Cav | −80 | 70 | 100 | 800 | 400 | 400 | 400 | 200 | 400 | 100 | 100 | 98 / 2800 |
| | KCa | −80 | 70 | 100 | 800 | 400 | 400 | 400 | 200 | 400 | 100 | 100 | 100 / 2800 |
| | Ih | −80 | 70 | 100 | 800 | 400 | 400 | 400 | 200 | 400 | 100 | 100 | 100 / 2800 |
| AP | Ion Type | $T_1$ | $T_A^*$ | $T_B^*$ | | | | | | | | | |
| | Kv | 1800 | 100 | 1800 | | | | | | | | | |
| | Nav | 1800 | 98 | 1800 | | | | | | | | | |
| | Cav | 1800 | 98 | 1800 | | | | | | | | | |
| | KCa | 1800 | 95 | 1655 | | | | | | | | | |
| | Ih | 1800 | 95 | 1655 | | | | | | | | | |

correlations across protocols were removed by again dimensionality-reducing by PCA (99% variance criterion) to yield a final score vector for each model. Since response traces were relatively noise-free, a high PCA dimensionality can be chosen to capture current response dimensions that are rare across the population of models. The precise value of the variance criterion in both PCA steps, although slightly changing the resulting scores, did not affect the clustering results reported above. The similarity between two channel models was then defined as the Euclidean distance of their dimensionality reduced scores.

In summary, the principal components calculated in the first step represented response curves along which the current traces were projected to yield intermediate scores. The second transformation was a linear mixing matrix, which combined these intermediate scores. Final scores had between 16 and 29 dimensions depending on the family analyzed, which additionally allowed the efficient storage of the characteristics of the thus compressed response properties in our resource. The linear PCA transformations, once calculated, can be applied to additional channel models and their current

responses and allow us to efficiently score new channels and easily evaluate them against all other channels in the resource (*ICGenealogy, 2016*).

## Clustering

For clustering of channel scores, we used Ward's minimal variance linkage (*Ward, 1963*) for hierarchical clustering, as implemented in the MATLAB Statistics Toolbox (R2015A, The MathWorks Inc., Natick, MA). This method can be used to produce a division of the set of all channel models into an arbitrary number of 'similar' clusters, the number of which has to be constrained by internal criteria (we assumed no a-priori existence of classes in this dataset) (*Halkidi et al., 2001*). To this end we employed a range of internal clustering evaluation measures, which indicate the emergence of an appropriate number of clusters. Although the evaluation of these measures requires some heuristics, they have been well established and can guide the decision as to which number of clusters to choose. Concretely, these are: the Silhouette criterion (*Rousseeuw, 1987*), the Dunn index (*Dunn, 1973*), the Davies-Bouldin index (*Davies and Bouldin, 1979*), and the Calinski-Harabasz measure (*Calinski and Harabasz, 1974*), also implemented in the MATLAB Statistics Toolbox (R2015A, The MathWorks Inc., Natick, MA). For the Dunn index, the Silhouette index and the Calinksi-Harabasz measure, high values indicate mostly compact and well-separated clusters. The Davies-Bouldin index also indicates compactness and separation, however for low values. For details and reviews on these clustering indexes see e.g., *Milligan and Cooper (1985)* and *Halkidi et al. (2001)*.

Values for the indexes and heuristics applied to arrive at the cluster numbers of the main text are given in *Figure 3—figure supplements 1* and *2*. Due to the natural partition of our dataset into five conditions used to calculate the final score, we also included a measure of heterogeneity computed directly on the traces of each condition, which we termed the 'Inner distance'. For a given condition, let $\mathbf{s}_j \in \mathbb{R}^{512 \cdot c}$ be the appended (over all possible voltage steps $c$) subsampled current responses, where $j \in \{1, \ldots, n_{\text{channels}}\}$ runs over all channels. Let $\{C_k | k \in \{1, \ldots, n_{\text{clusters}}\}\}$ be a clustering of all channels – a collection of sets, such that each channel index $j$ is contained in a single set. Let $\mathbf{c}_k = \frac{1}{|C_k|} \sum_{j \in C_k} \mathbf{s}_j$ be the mean response trace of each cluster. The inner distance is then calculated as the scatter around the mean, averaged over all clusters:

$$d_{\text{inner}} = \frac{1}{n_{\text{clusters}}} \sum_k \frac{1}{|C_k|} \sum_{j \in C_k} \|\mathbf{s}_k - \mathbf{c}_k\|. \tag{1}$$

To make the measure comparable across different conditions, which might have different values of $c$ (the number of voltage steps), we define the norm as $\|\mathbf{s}_j\| = \frac{1}{c*512} \sum_{i=1}^{c*512} |s_j(i)|$.

A number of additional linkage methods (complete, single, average) and metrics (cityblock, squared Euclidean) were also evaluated. While giving comparable performance on a synthetic test set, they yield mostly inferior subsections of the full set of channels with very high numbers of single elements being isolated as separate clusters.

## Assessment of protocols

To qualitatively assess the necessity of the voltage-clamp protocols for separation of labeled subtypes, the condition scores of all channel models of a particular subtype were compared with those of other subtypes (*Figure 3—figure supplement 4*). We show that certain protocols are more important for differentiating particular subtypes: for example, Kv models of the *m-type* show a large distinction from *A-type*, *dr* and *HH* subtypes in the condition scores of the action potential protocol, whereas *A-type* channel models show distinct condition scores in the activation, inactivation and deactivation protocols. The protocols chosen here thus exploit a necessary range of response kinetics; the general method of deriving a final score from each of the conditions, however, is amenable to straight-forward extension by further protocols or second-order features extracted from the response traces, as for example peak response values and time-scales (*Lewicki, 1998*; *Druckmann et al., 2013*). Each of these could be incorporated in the analysis as additional condition scores.

An alternative for the characterization of ion channel dynamics would be given by a model-based approach: by fitting the parameters of a single *super-model* to closely approximate the dynamics of all other channel models on hand, one could characterize channel models by the resulting parameter

values. However, we could not assume that such a single super-model would capture the full dynamical diversity we were presented with in our dataset. Similarly, the diverse kinetics of biological ion channels measured in experiments might not be captured adequately by a single super-model. We thus chose a standardized 'model-free' approach, which can be extended (see above) should the need for more detailed discrimination of channel kinetics (from model and experiment alike) arise.

## Generation of circos plots

'Circos' plots displaying the clustering results together with genealogical links were generated using the Circos visualization tool (*Krzywinski et al., 2009*), combined with TreeDyn (*Chevenet et al., 2006*) to create circular dendrograms (*Figure 4D*).

The Circos plot is a visualization technique that enables the comparison of functional similarity to metadata information for each cluster separately. All *unique* models of a given class are arranged in a circle (numbers of duplicate models are shown in the gray histogram along the circle). *Plot rings*: For each model, the following information is shown (from inside to outside; refer to legend): clustering dendrogram, subtype label, histogram count of duplicate models, model runtime information, number of citations of the accompanying paper, and most common subtype label(s) of each cluster (all subtypes that contribute $30\%$ or more). Location along the circle corresponds to functional (clustering) information, whereas color corresponds to metadata information, such as subtype label. *Location*: position along the circle was established by the circular dendrogram at the center. This dendrogram was created with an agglomerative hierarchical clustering algorithm as defined above, and arranges models in such a way that similar models are in adjacency, and all models in the same cluster appear in one continuous group. The outer ring of the plot denotes the extent of each individual cluster. Groups of models defined by cluster were visually displaced from others by adding a small white space between clusters. *Color*: Three color legends accompanying the graph define the color relationships plotted. The two large rings on the inside and outside are colored by subtype label (of individual models and of clusters, respectively), of which the 11 most common are displayed, with all others in gray. Two smaller rings just inside of the outer ring, denoting the number of citations and runtime, are colored on a red-blue scale.

## Generation of genealogy figures

Pedigree plots were generated using Gephi (*Bastian et al., 2009*), and then manipulated and ordered manually for visualization (*Figure 2E*, *Figure 4A*). Coloring was chosen according to subtype label as well as cluster identity. 'Sankey' diagrams (*Sankey, 1898*; *Schmidt, 2008*) were generated in Javascript and D3.js (*Bostock et al., 2011*) (*Figure 4B*, *Figure 5B*). Subtype coloring was chosen as for the pedigree plots. Subtype labels, clusters and families were arranged from top to bottom by size.

All other figures were generated using MATLAB (R2015A, The MathWorks Inc., Natick, MA) and Python 2.7 with matplotlib 1.4.2.

## Relational database, API and web interface

All collected metadata, as well as final scores and clustering results were organized in a relational MYSQL database, which is openly queryable through a web API. Details about database structure and the implementations of the web-application and API will be made available on the website (*ICGenealogy, 2016*). The graphical channel browser frontend was developed in Javascript and D3.js (*Bostock et al., 2011*) by Phyramid Ltd, Bucharest, Romania.

## Code availability

Code for the generation of current response traces in *NEURON* as well as for the analysis of current traces will be made available on the website (*ICGenealogy, 2016*).

## Electrophysiology

K+ currents were recorded from Drosophila Kenyon cells in targeted in vivo whole-cell voltage clamp experiments as previously described (*Murthy and Turner, 2013*). Male NP7175-GAL4;UAS-mCD8-GFP flies were immobilized and fixed to a perfusion chamber using wax. Cuticle, adipose tissue, trachea and perineural sheath were removed in a window large enough to expose the posterior brain.

The preparation was continuously superfused with extracellular solution containing (in mM) 103 NaCl, 3 KCl, 26 NaHCO$_3$, 1 NaH$_2$PO$_4$, 1.5 CaCl$_2$, 4 MgCl$_2$, 5 TES, eight trehalose, 10 glucose and seven sucrose (pH 7.3 when equilibrated with 5% CO$_2$ and 95% O$_2$). Tetrodotoxin was added at a final concentration of $1\,\mu M$. Borosilicate glass electrodes (14–16 M$\Omega$) were filled with pipette solution containing (in mM) 140 potassium aspartate, 1 KCl, 10 HEPES, 4 MgATP, 0.5 Na$_3$GTP and 1 EGTA (pH 7.3). All experiments were performed at room temperature ($21 - 23\,°C$). Signals were recorded with a MultiClamp 700B Microelectrode Amplifier, lowpass-filtered at 10 and digitized at 50 kHz using a Digidata 1440A digitizer controlled via the pCLAMP 10 software (all Molecular Devices). Capacitive transients and linear leak currents were subtracted using a P/4 protocol and all traces were corrected for the liquid junction potential (*Neher, 1991*). Voltage pulse protocols were applied as indicated for Kv (*Figure 3—figure supplement 1*; *Table 2*) and data were analyzed in MATLAB. Resulting current traces were processed analogously to model current traces, as specified in section *Data extraction and processing*.

## Acknowledgements

The channel browser web-frontend was created by Phyramid (www.phyramid.com). We thank V Harbuz and P Toader from Phyramid for helpful discussions about visualization, as well as P Gleeson, S Crook, J Kwag, NA Cayco-Gajic and Y Poirazi for valuable comments. We thank S Bailey and L Phillimore for data collection that has contributed to the continued maintenance of the database and website. We also thank the reviewers for several helpful suggestions that have made this work more clear and complete. Research was supported by grants from the Wellcome Trust Research Fellowship (GM and TPV). AS was supported by the Swiss National Science Foundation (200020_147200). RR was supported by the EPFL Blue Brain Project Fund and ETH Board funding to the Blue Brain Project. LNG was supported by a Wellcome Trust OXION studentship.

## Additional information

### Funding

| Funder | Grant reference number | Author |
| --- | --- | --- |
| Wellcome | WT100000 | William F Podlaski<br>Tim Vogels |
| Schweizerischer Nationalfonds zur Förderung der Wissenschaftlichen Forschung | 200020_147200 | Alexander Seeholzer |
| Wellcome | OXION | Lukas N Groschner |
| Wellcome | WT090309 | Gero Miesenböck |
| Wellcome | WT106988 | Gero Miesenböck |
| EPFL Blue Brain Project Fund | | Rajnish Ranjan |
| ETH Board funding to the Blue Brain Project | | Rajnish Ranjan |

The funders had no role in study design, data collection and interpretation, or the decision to submit the work for publication.

### Author contributions

WFP, AS, Conceptualization, Data curation, Software, Formal analysis, Investigation, Visualization, Methodology, Writing—original draft, Writing—review and editing; LNG, GM, Investigation, Methodology, Writing—original draft, Writing—review and editing; RR, Conceptualization, Data curation, Software, Formal analysis, Supervision, Investigation, Visualization, Methodology, Writing—original draft; TPV, Conceptualization, Formal analysis, Supervision, Funding acquisition, Investigation, Methodology, Writing—original draft, Writing—review and editing

Author ORCIDs
William F Podlaski, http://orcid.org/0000-0001-6619-7502
Alexander Seeholzer, http://orcid.org/0000-0002-1541-4906
Lukas N Groschner, http://orcid.org/0000-0002-4325-0232

## Additional files

### Supplementary files

• Supplementary file 1. Instruction manual for ICGenealogy website browser. This file contains a brief tutorial which introduces all of the major aspects of the accompanying web browser for the database.

• Supplementary file 2. Supplementary references. This file lists all model files contained in the database and used for analysis here, as well as the accompanying reference publication for each one.

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
