## [Decision Letter]

Thank you for submitting your article "ICGenealogy: Mapping the function of neuronal ion channels in model and experiment" for consideration by *eLife*. Your article has been favorably evaluated by Eve Marder (Senior Editor) and three reviewers, one of whom, Frances Skinner, is a member of our Board of Reviewing Editors. The following individuals involved in review of your submission have agreed to reveal their identity: Timothy O'Leary (Reviewer #2); Farzan Nadim (Reviewer #3).

The reviewers have discussed the reviews with one another and the Reviewing Editor has drafted this decision to help you prepare a revised submission.

Summary of the work

For many years the physiological community has quantified ion channel kinetics for the purpose of building models, and for understanding the biological importance of the diversity of ion channels expressed in neurons and other cells. However, until now, there has not been a useable and comprehensive resource for documenting, organising and comparing ion channel models. The current manuscript offers a new, comprehensive resource for channel models, ICGenealogy, as well as analysis tools for comparing channel models.

ICGenealogy allows access to a large pool of already publicly available information regarding channel models (i.e., ion channel models in NEURON software as deposited into ModelDB), and classifies them through a genealogy method. It uses a standard, if somewhat ad hoc, set of protocols to classify the currents and uses PCA and a metric to classify them and explore their "distance" to build clusters. It facilitates the viewing of model channel development metadata as well as allows standardized comparisons of ion channel models. This would otherwise be time-consuming and inefficient for individual modellers to do on a per model basis. It thus offers much needed standardization to the field of model development.

Further, and most importantly, ICGenealogy facilitates model comparisons with experimental data so that more informed decisions can be made when using particular channel types in a model.

In summary, ICGenealogy provides diligence for ion channel modeling and should serve as a most helpful tool to bring model and experiment together in terms of perspectives and transparency of assumptions and rationales. Overall, all reviewers agreed that ICGenealogy is a useful and important contribution to the field.

Essential revisions:

1) Biological variability:

The issue of biological variability is touched on (subsection “Ion channel model groups defined by common metadata show variability in behavior”, first paragraph), but more should be said about this. At present, it is discussed too passively.

Even a single ion channel gene has multiple splice variants, subunit combinations, and possibilities for postranslational modification. Computationally-minded readers can fail to appreciate this and write-off the biological variation as noise or experimental incompetence. This is exacerbated by the subsequent assertions:

"A large portion of the variability may also stem from model fitting and idiosyncratic changes to individual model implementations. Consistent with this notion, we find that models defined by common families can occasionally fall into different clusters (Figure 4). This suggests that experimental data used to fit models can be treated in different ways, perhaps combined with other data, and may lead to disparate models."

Actually, if you experimentally isolate a current pharmacologically, or even go to the trouble of cloning a channel gene to express it heterologously, you may still see measurable differences in kinetics within an 'identified channel type', for the reasons above. It is thus important to state more clearly that this variability might be genuine, even if we find it a nuisance and don't fully understand its possible significance.

Further, ion channels can have tens of phosphorylation sites that could allow for activity or neuromodulation to change their activation/inactivation properties. For example, Misonou et al., "Regulation of ion channel localization and phosphorylation by neuronal activity." Nat Neurosci 2004.

2) Maintenance and usage of resource:

As with all resources of this kind, its usefulness depends on its continued maintenance. What plans do the authors have in mind to achieve this? This should be delineated in the manuscript in some way.

In other words, the authors should take steps to ensure the maintenance of this resource is self-sustaining. It is only useful insofar as the community continues to use, update and build on its features.

The authors have constructed the resource to allow upload of data, but what capacity is there for ongoing, community-based curation? For example, can existing models be queried and annotated, or cross-referenced to external resources (such as EBI, NCBI, Allen Brain Atlas…)? These features would contribute to the longevity and flexibility of the resource and prevent a fate common to other resources that fade over time. It would be useful for a clear path to these additions to be laid out (e.g. in API documentation), even if, as the authors emphasize, the actual implementation is beyond the scope of the study.

I really loved the comparison of the *Drosophila* K^+^ current with existing models. Would be to allow submission of such experimental data on ionic currents, recorded with the standardized protocol of ICG, to the site so that best-matching models and geneology could be identified. This would greatly increase the value of this resource.

3) Potentially confusing terminology:

To avoid confusion and ambiguity, it would be helpful if the author specifically say 'ion channel model' or 'neuronal/cell model' rather than just 'model' in various places throughout the paper (e.g., Introduction, end of first paragraph, and several subsequent places).

Throughout the manuscript the authors should be more careful to distinguish the sense in which 'genealogy' is used. There is clear scope for confusing 'model genealogy' (i.e. similarity in behaviour), 'publication genealogy' (i.e. where the model was first described and its subsequent uses and citations) and 'genetic genealogy' in the biological sense! I think the authors should define their terms carefully at the beginning of the manuscript and occasionally remind readers of the differences.

---

## [Author Response]

Essential revisions:

1) Biological variability:

The issue of biological variability is touched on (subsection “Ion channel model groups defined by common metadata show variability in behavior”, first paragraph), but more should be said about this. At present, it is discussed too passively.

Even a single ion channel gene has multiple splice variants, subunit combinations, and possibilities for postranslational modification. Computationally-minded readers can fail to appreciate this and write-off the biological variation as noise or experimental incompetence. This is exacerbated by the subsequent assertions: "A large portion of the variability may also stem from model fitting and idiosyncratic changes to individual model implementations. Consistent with this notion, we find that models defined by common families can occasionally fall into different clusters (Figure 4). This suggests that experimental data used to fit models can be treated in different ways, perhaps combined with other data, and may lead to disparate models."

Actually, if you experimentally isolate a current pharmacologically, or even go to the trouble of cloning a channel gene to express it heterologously, you may still see measurable differences in kinetics within an 'identified channel type', for the reasons above. It is thus important to state more clearly that this variability might be genuine, even if we find it a nuisance and don't fully understand its possible significance.

Further, ion channels can have tens of phosphorylation sites that could allow for activity or neuromodulation to change their activation/inactivation properties. For example, Misonou et al., "Regulation of ion channel localization and phosphorylation by neuronal activity." Nat Neurosci 2004.

We thank the reviewers for pointing out how important an issue this is, and acknowledge that we were not clear enough in our treatment of it. We have now modified the corresponding sections in the Results, as well as the Discussion, to focus on the aspects of true biological variability. The suggested reference has been added, along with several others that highlight the nature of ion channel variability. Additionally, we cite previous modelling work that has explored how ion channel variability manifests itself at the single neuron level.

While we still mention the possibility of increased diversity being caused by the experimental setup and model fitting, we hope to have clarified that this is secondary compared to the known variability of ion channels that exists in biological neurons. In addition, we mention in the Discussion a possible follow-up to this work, that involves studying the co-variability of sets of ion channel models together. We have secured a (BBSRC) grant that covers 6 postdoc-years and 3 technician-years for this extension (see below).

2) Maintenance and usage of resource:

As with all resources of this kind, its usefulness depends on its continued maintenance. What plans do the authors have in mind to achieve this? This should be delineated in the manuscript in some way.

In other words, the authors should take steps to ensure the maintenance of this resource is self-sustaining. It is only useful insofar as the community continues to use, update and build on its features.

We have extensive plans for the continued maintenance and improvement of the resource. Recently acquired funding through a BBSRC Grant (BB/N019512/1) for 3 years will place 2 post-doctoral researchers as well as 1 technician in charge of developing the features of the resource, as well as curating and updating the existing database.

One important component of this maintenance is updating the website with new models as they are published on ModelDB. Since the reviewer’s comments have been received, we have added over 500 new ion channel models to the database and website (confer https://icg.neurotheory.ox.ac.uk/development#content).

This recent extensive content update involved the work of two lab technicians who have now been added as tentative contributing authors to the manuscript (indicated by parentheses).

The state of the database and website as used to generate the figures in the submission have been conserved and made publicly available in a “Publication State Mirror” (linked from https://icg.neurotheory.ox.ac.uk/about and https://icg.neurotheory.ox.ac.uk/development).

The authors have constructed the resource to allow upload of data, but what capacity is there for ongoing, community-based curation? For example, can existing models be queried and annotated, or cross-referenced to external resources (such as EBI, NCBI, Allen Brain Atlas…)? These features would contribute to the longevity and flexibility of the resource and prevent a fate common to other resources that fade over time. It would be useful for a clear path to these additions to be laid out (e.g. in API documentation), even if, as the authors emphasize, the actual implementation is beyond the scope of the study.

The tight integration of our resource with other projects and external resources will indeed be a major goal for our development team. Currently, community-based curation is possible via a model-specific issue tracker (https://icg.neurotheory.ox.ac.uk/contribute).

We are planning to extend the existing model submission functionality to a degree that enables users to propose changes to models directly in the interface, as well as submit new models via our website.

As suggested by the reviewer, we have added a “Development” section in the website (https://icg.neurotheory.ox.ac.uk/development), where a timeline of upcoming features (including the above) will be maintained. Amongst others, we there point out the upcoming additions of “Generalized crossreferencing” (with resources besides ModelDB) and user contributions.

I really loved the comparison of the Drosophila K^+^ current with existing models. Would be to allow submission of such experimental data on ionic currents, recorded with the standardized protocol of ICG, to the site so that best-matching models and geneology could be identified. This would greatly increase the value of this resource.

We thank the reviewer for the encouraging comment. Indeed uploading of experimental data recorded with the ICG protocols is already possible at present. This is demonstrated, for example, in the instructional video on our website, and explained a little more detailed in the “Submit a model” section (https://icg.neurotheory.ox.ac.uk/submit).

Additionally, as part of the BBSRC grant mentioned above, we are planning to integrate experimentally acquired datasets recorded with the same protocols into our database to allow the direct comparison of experiments to models in our web resource. We plan to make standardized ICG voltage-clamp protocols available for some of the most common data acquisition programs such as pCLAMP (Molecular Devices), Pulse (HKEA), Igor Pro (WaveMetrics), and Signal (CED).

3) Potentially confusing terminology:

To avoid confusion and ambiguity, it would be helpful if the author specifically say 'ion channel model' or 'neuronal/cell model' rather than just 'model' in various places throughout the paper (e.g., Introduction, end of first paragraph, and several subsequent places).

We agree, and have now gone through the text in order to clarify all usage of the word ‘model’ when appropriate.

Throughout the manuscript the authors should be more careful to distinguish the sense in which 'genealogy' is used. There is clear scope for confusing 'model genealogy' (i.e. similarity in behaviour), 'publication genealogy' (i.e. where the model was first described and its subsequent uses and citations) and 'genetic genealogy' in the biological sense! I think the authors should define their terms carefully at the beginning of the manuscript and occasionally remind readers of the differences.

We acknowledge that this may be confusing for many readers. We have now added an extra statement in the Results (when introducing the ancestor-descendant relationship), as well as a full paragraph in the Methods which clarifies how the word genealogy is used throughout the text.